# Spin-momentum locking and spin-orbit torques in magnetic nano-heterojunctions composed of Weyl semimetal WTe$_2$

Peng Li [1], Weikang Wu[2], Yan Wen[1], Chenhui Zhang [1], Junwei Zhang[1], Senfu Zhang [1], Zhiming Yu [2], Shengyuan A. Yang [2], A. Manchon [1] & Xi-xiang Zhang [1]

Spin–orbit torque has recently been intensively investigated for the purposes of manipulating the magnetization in magnetic nano-devices and understanding fundamental physics. Therefore, the search for novel materials or material combinations that exhibit a strong enough spin-torque effect has become one of the top priorities in this field of spintronics. Weyl semimetal, a new topological material that features open Fermi arc with strong spin–orbit coupling and spin–momentum locking effect, is naturally expected to exhibit an enhanced spin-torque effect in magnetic nano-devices. Here we observe a significantly enhanced spin conductivity, which is associated with the field-like torque at low temperatures. The enhancement is obtained in the b-axis WTe$_2$/Py bilayers of nano-devices but not observed in the a-axis of WTe$_2$/Py nano-devices, which can be ascribed to the enhanced spin accumulation by the spin–momentum locking effect of the Fermi arcs of the Weyl semimetal WTe$_2$.

---

[1] King Abdullah University of Science and Technology, Physical Science and Engineering Division, Thuwal 23955-6900, Saudi Arabia. [2] Research Laboratory for Quantum Materials, Singapore University of Technology and Design, Singapore 487372, Singapore. These authors contributed equally: Peng Li, Weikang Wu, Yan Wen. Correspondence and requests for materials should be addressed to X.-X.Z. (email: xixiang.zhang@kaust.edu.sa)

The effect of spin–orbit torques has been extensively explored with regard to the switching of the magnetization in the magnetic layer through the transferring of angular momentum carried by the spin current in different magnetic nano-heterostructures. It has therefore been considered a promising technology in terms of its applications to computation, logic, and memory[1–3]. Owing to the existence of strong spin–orbit coupling (SOC) in topological materials[4,5], such as topological insulators, the magnetic heterostructures composed of topological materials have received significant attention for both their potential to advance the fundamental understanding of the underlying physics and possible applications[6–8]. It has been shown that the spin–momentum locking in the surface states of the topological insulators is able to induce a non-equilibrium spin accumulation into the adjacent magnetic layer that switches and manipulates the magnetization of the magnetic layer[6,7,9–13]. Furthermore, the spin accumulation in the topological surface states can be electrically detected by measuring the hysteresis loops of the in-plane resistance of the magnetic tunneling junction devices[14–17].

Recently, three-dimensional (3D) Dirac and Weyl topological semimetals have been acknowledged as a new state of topological quantum matter with a linear dispersion at the Dirac or Weyl points[18–27]. A number of intriguing properties have been observed in these topological semimetals, such as high carrier mobility, extremely large magnetoresistance (MR), and, especially, the existence of surface states (i.e., the Fermi arcs at the surfaces of the sample). These Fermi arcs are robust against scattering, which should lead to interesting properties in the magnetic heterostructures composed of Dirac and Weyl topological semimetals.

It has been predicted and then experimentally demonstrated that WTe$_2$ is a type-II Weyl semimetal[28–30] in which the Weyl points occur at the crossing of the oblique conduction and the valence bands due to the broken inversion symmetry[23]. The calculations demonstrate that the Weyl points in the topological Weyl semimetal WTe$_2$ will sustain up to approximately 100 K[30]. Above approximately 100 K, the band crossing (Weyl points) will separate, and WTe$_2$ will transform into a conventional semimetal due to the slight increase in lattice constants. An important feature of Weyl semimetals is the existence of topological surface states, namely Fermi arcs. These open Fermi arcs connect the projections of the bulk Weyl points of the opposite chiralities on the surfaces of the sample to form extra electrical conduction loops (i.e., Weyl orbits)[28,31,32]. Recent studies have demonstrated that the Fermi arcs of WTe$_2$ are approximately situated along the **Y** due to the broken inversion symmetry[33,34]. More importantly, it has been suggested that the Fermi arc in Weyl semimetals are spin-polarized with similar spin–momentum locking effects as have been observed in the surface states of topological insulators[35–38]. Although an out-of-plane damping-like (DL) torque induced by broken inversion symmetry was observed in the WTe$_2$/Py heterostructures at room temperature[26,27], the spin–orbit torques originating from the topological Fermi arc remain undiscovered.

In this work, we electrically detect the spin-polarized surface states in the WTe$_2$/Al$_2$O$_3$/Fe tunneling junctions and extract the spin–orbit torques of WTe$_2$/Py devices using the second-harmonic measurements. The spin–momentum locking of Fermi arc is further demonstrated by the hysteresis loops of the resistance observed in the electrical measurements of the WTe$_2$/Al$_2$O$_3$/Fe tunneling junction. We observe the switching between high and low tunnel-resistance states upon sweeping the magnetic field along a direction parallel to the accumulated spin of the surface states in the WTe$_2$ ribbons. We also demonstrate that the topological Fermi arcs of WTe$_2$ are along the **Y**-direction (b-axis)

and that the spin–momentum locking effect enhances the field-like (FL) torques of the WTe$_2$/Py bilayers at low temperatures.

## Results

**Two-dimensional transport in WTe$_2$/Py heterostructures.** In our previous work, we observed that a negative magnetoresistance (NMR) was induced by the chiral anomaly and that an extra quantum oscillation along the b-axis of WTe$_2$ ribbon was induced by Fermi arcs[28]. Surprisingly, we are still able to observe the trace of NMR in WTe$_2$/Py bilayers, when both the magnetic field and electric field are applied along the b-axis (Fig. 1a); though the critical magnetic field shifted to approximately 12 T from 4.8 T in pure WTe$_2$, as displayed in the inset of Fig. 1a. This much higher critical magnetic field observed in WTe$_2$/Py bilayers may be caused by the slight shift in the Fermi levels locating far from the Weyl points of the semimetal, which could be associated with the proximity effect of the ferromagnetic layer on WTe$_2$. Therefore, the NMR observed in the WTe$_2$ (20 nm)/Py (6 nm) bilayers in the high magnetic field should be caused by a chiral anomaly in WTe$_2$ (inset of Fig. 1a). We also observed the extra quantum oscillation frequency in the spectrum of the fast Fourier transform (FFT) of the MR data of WTe$_2$/Py bilayers (**H**//c). As shown in Fig. 1b, this extra peak is the same as that observed in WTe$_2$ ribbons and should originate from the Weyl orbit formed by Fermi arcs[28,31,32]. The inset of Fig. 1b depicts the data obtained at 2 K from the WTe$_2$/Py bilayers that exhibit the strong Shubnikov de Haas (SdH) oscillations. The existence of NMR and Weyl orbit quantum oscillations in WTe$_2$/Py bilayers strongly suggests that WTe$_2$ retains as a Weyl semimetal in the bilayer.

To understand the underlying physics, we measured the angular-dependent magnetoresistance (AMR) on a WTe$_2$ ribbon with a thickness of $t = 20$ nm at 2 K by varying the angle $\theta$. We found that the AMR follows the $\cos2\theta$ dependence (Fig. 1c), indicating a 3D bulk feature of WTe$_2$, which agrees with our previous work[28]. Interestingly, the angular dependence of the AMR obtained on the WTe$_2$/Py bilayer at 2 K deviates significantly from the $\cos2\theta$ dependence; however, it can be well described by a $|\cos\theta|$ dependence (Fig. 1d), indicative of two-dimensional (2D) conduction in the WTe$_2$/Py bilayer[39,40] To explore the origin of the 2D transport, we measured AMR in Py film of 6 nm thick as we did for the WTe$_2$ ribbon of $t = 20$ nm. As expected, the AMR obtained from Py film can also be well described by $\cos2\theta$ dependence (Supplementary Fig. 1). The observation of 3D transport in both individual WTe$_2$ and Py film strongly indicates that the $|\cos\theta|$ dependence observed in the WTe$_2$/Py bilayer is from neither the WTe$_2$ ribbon nor the Py film. We thus believe that the 2D magneto-transport characteristic should originate from the conduction channel formed at the interface between the WTe$_2$ and Py[39,40]. The significantly sharp and high-quality interface between WTe$_2$ and Py, along with the top surface of WTe$_2$, is clearly observed in the high-resolution transmission electron microscopy image of the cross-section of the WTe$_2$/Py bilayer (inset of Fig. 1d and Supplementary Fig. 2), suggesting that the 2D feature of the transport of the Fermi arc electrons may remain unchanged. Above 100 K, with increasing temperature, the 2D feature of the AMR in WTe$_2$/Py bilayers gradually vanishes (Supplementary Fig. 3). To further uncover the origin of quasi-2D magnetotransport in WTe$_2$/Py bilayers, we fabricated a device of WTe$_2$ (20 nm)/Au (5 nm) bilayers and measured the angular dependence of its AMR. Interestingly, we observed the same AMR angular dependence of AMR on $|\cos\theta|$ as in WTe$_2$ (20 nm)/Py (6 nm) bilayers (Supplementary Fig. 3). We then calculated the density of states (DOS) of the interface near the Fermi level in WTe$_2$/Au. As compared to that in a bulk WTe$_2$, the DOS of WTe$_2$ near the WTe$_2$/Au interface is nearly doubled

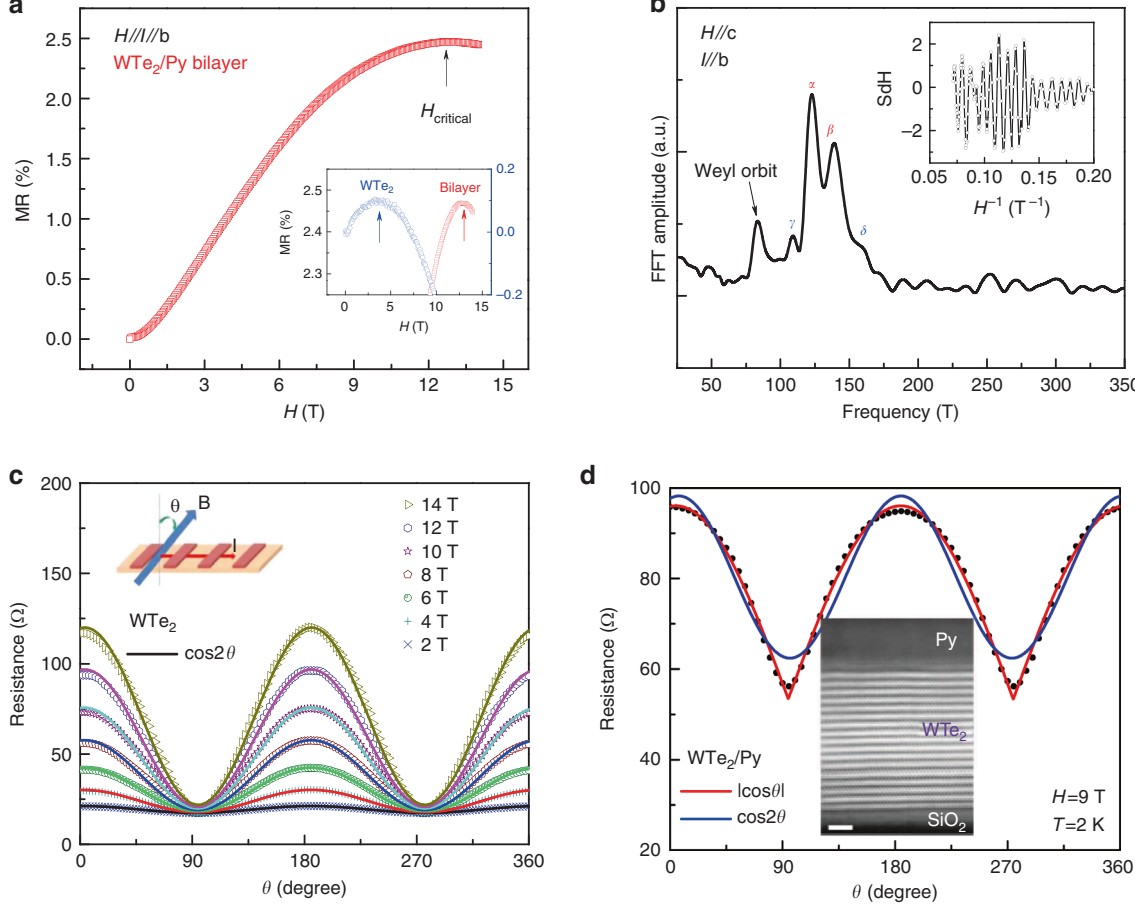

**Fig. 1** Two-dimensional electronic transport and Weyl features in WTe$_2$/Py bilayers. **a** The magnetoresistance measured on a WTe$_2$/Py bilayer at 2 K with the **I**//**H**//$b$-axis of WTe$_2$. Chiral anomaly induced negative longitudinal magnetoresistance appeared above 12 T. The blue and red symbols in the inset represent the amplification of the field dependence of the longitudinal magnetoresistance of WTe$_2$ and WTe$_2$/Py bilayers, measured at 2 K with the **I**//**H**// $b$-axis of WTe$_2$. **b** The FFT spectrum of the SdH oscillation of the $b$-axis WTe$_2$/Py bilayers. A frequency of approximately 80 T induced by Weyl orbit oscillation appeared in addition to the peak from the bulk WTe$_2$. The inset shows the raw data of the SdH oscillation extracted from the longitudinal magnetoresistance data. **c** The angular-dependent magnetoresistance of the WTe$_2$ ($t = 20$ nm) under different magnetic fields. All of the data can be well described by the $\cos 2\theta$ dependence, a feature of 3D transport. The inset schematically presents the configuration of the measurement. **d** The angular-dependent magnetoresistance of the WTe$_2$/Py bilayers ($T = 2$ K and $H = 9$ T). The data are more adequately described by a $|\cos \theta|$ function than by $\cos 2\theta$, suggesting a 2D transport mechanism in the device rather than a 3D transport mechanism. The inset contains a high-angle annular dark-field cross-sectional image of WTe$_2$/Py, indicating the high quality of the layers and sharp interface. The white scale bar is 2 nm

(Supplementary Fig. 4), which may account for the quasi-2D electron transport in WTe$_2$/Py and WTe$_2$/Au. The observation of the 2D nature of the electron transport in both bilayers (WTe$_2$/Py and WTe$_2$/Au) suggests that this 2D transport may be closely associated with the topological surface state of the mechanically exfoliated (001)-oriented WTe$_2$[23]. Since the topological surface states are present at the top and bottom surface of the mechanically exfoliated (001)-oriented WTe$_2$[23], it is therefore intriguing to investigate WTe$_2$ to determine whether there is a spin–momentum locking effect in its surface states and whether the effect of spin–momentum locking will enhance the spin torque in the magnetic heterostructures.

**Electrical detection of the surface states**. The electrical detection of the spin–momentum locking effect of surface states has been realized even at room temperatures in ferromagnetic tunnel junctions composed of topological insulators[14–17,41]. To examine the effect of the spin–momentum locking in topological Weyl semimetals, we then fabricated the ferromagnetic tunnel junctions WTe$_2$ ($a$- and $b$-axis)/Al$_2$O$_3$/Fe (Fig. 2a and Supplementary

Fig. 5) and measured the tunneling resistance by varying external magnetic field and applying a constant DC electric current.

Before analyzing the experimental results, we must first discuss the theoretical results. Since the Fermi arcs are usually extended in the surface Brillouin zone and their shapes are less constrained, the spin texture for the arc states generally cannot be described by a simple local Hamiltonian. However, the spin–momentum-locked spin textures can be revealed by first-principles calculations[12,13] and mapped out in experiment. Our calculation result for the Fermi arc spin texture is shown in Fig. 2b. Figure 2b depicts the calculated correlation between the momentum and spin direction of electrons in the surface states of WTe$_2$ (Supplementary Fig. 6). The arrows indicate the direction of spin. Several key features should be noted. First, eight Weyl points locate at the crossing of electron and hole pockets. Second, four open Fermi arcs connecting the Weyl points with opposite chiralities in each quadrant are quite small and along the **Y**-direction. Third, four conventional surface states in each quadrant end into bulk electron Fermi pockets and two conventional surface states connect the Weyl points in different quadrants. For topological insulators, the spin-polarized transport

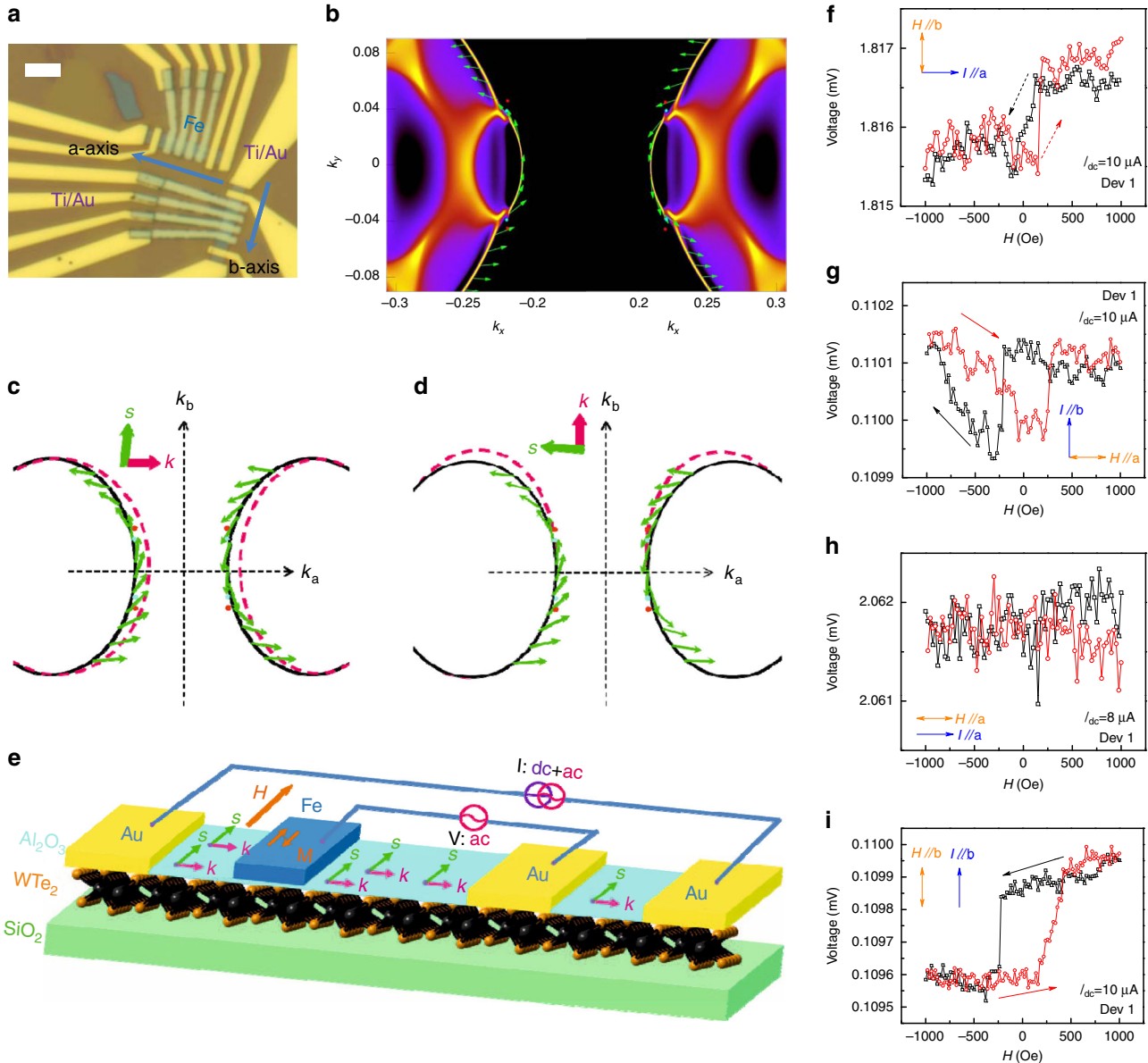

**Fig. 2** Spin–momentum locking in WTe$_2$/Al$_2$O$_3$/Fe tunnel junctions. **a** An optical image of a WTe$_2$(14 nm)/Al$_2$O$_3$(3 nm)/Fe(6 nm) tunnel junction device. The arrows indicate the in-plane crystal orientation of WTe$_2$. The scale bar is 5 μm. **b** A schematic illustration of the relationship between the spin and the momentum of electrons in the surface states in WTe$_2$. The red and cyan spots represent the Weyl points with opposite chiralities. The green arrows represent the spin orientation of electrons in surface states. **c, d** A schematic of spin–momentum locking in the conventional surface states of WTe$_2$ under different electric fields: **E**//+x and **E**//+y, respectively. The dashed purple line signifies the surface states after applying an electric field. The spins are approximately orthogonal to the momentum. A schematic illustration of the devices and the corresponding measurement configuration of the electrical detection of the spin–momentum locking of the surface states in WTe$_2$. A DC current of 8–50 μA and an AC current of 1 μA (lock-in method) were applied during the measurement. Under the application of the DC current, the spin (**s**) of the electrons in the surface states is accumulated, with the direction orthogonal to their momentum (**k**). Therefore, a low resistance is detected when the direction of the magnetic momentum of Fe is parallel to the accumulated spin magnetic moment, and vice versa. The momentum of the ferromagnetic layer is switched by an in-plane magnetic field. **f–i** The voltage measured across two inner electrodes for the tunnel junction (WTe$_2$/Al$_2$O$_3$/Fe) as a function of in-plane magnetic field. The configurations between the DC current direction and the magnetic field direction are indicated in the figures ($T = 2$ K). The configurations in different panels are **f**: **I$_{dc}$**//a ($I_{dc} = 8$ μA), **H**//b; **g**: **I$_{dc}$**//b ($I_{dc} = 10$ μA), **H**//a; **h**: **I$_{dc}$**//a ($I_{dc} = 10$ μA), **H**//a; and **i**: **I$_{dc}$**//b ($I_{dc} = 10$ μA), **H**//b. The thickness of the WTe$_2$ in this device (Device 1) is 20.0 nm

from the topological surface states can be understood as a result of the spin Hall effect in the bulk, establishing the bulk boundary correspondence[42]. Given that a Weyl semimetal may be regarded as the limiting case of a topological insulator (with the gap approaching zero) and it usually exhibits strong spin Hall effect[42], a similar picture also applies here. Namely, under an applied **E** field, the resulting transverse pure spin current in the bulk of a

Weyl semimetal leads to opposite spin accumulations on the top and bottom surfaces, generating surface spin-polarized charge currents according to the Fermi arc spin textures. To present clearly the direction of the spins accumulated in the surface states under the electric field, we schematically illustrate the surface states in Fig. 2c, d. The dashed pink lines indicate the locations of the surface states under the electric field **E**. The accumulated spins

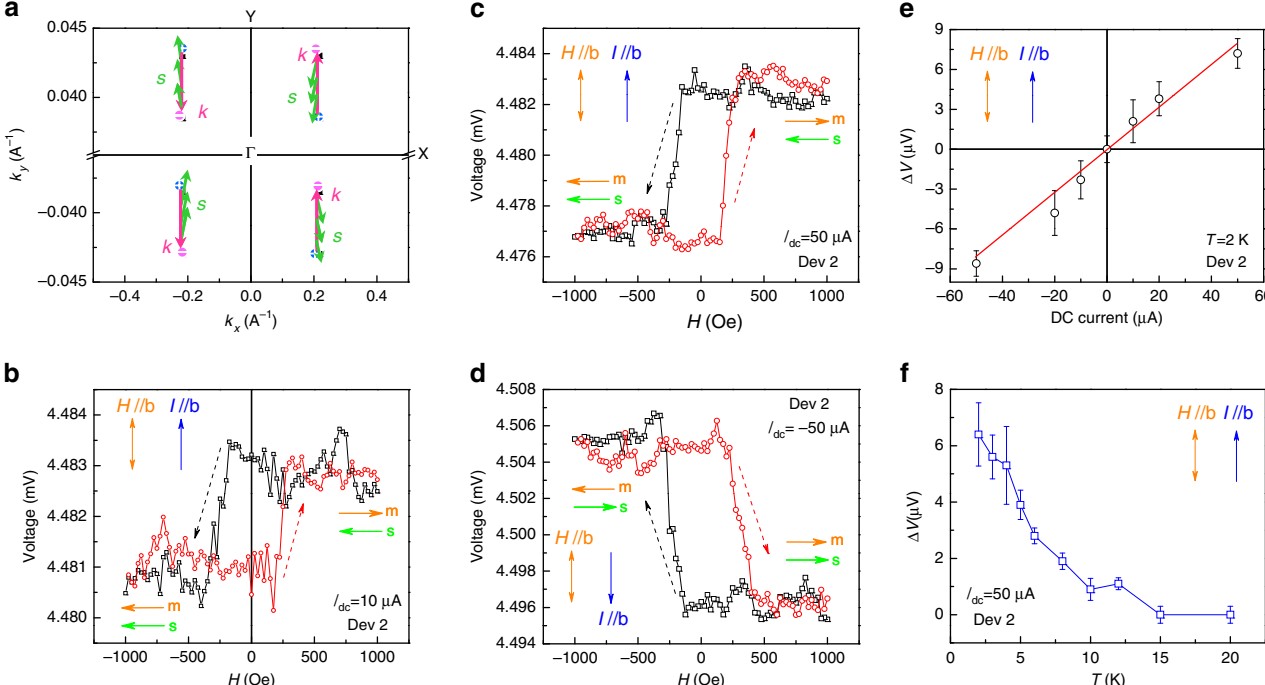

**Fig. 3** Voltage hysteresis loop in b-axis WTe$_2$/Al$_2$O$_3$/Fe tunnel junctions. **a** schematic of the spin–momentum locking of the topological Fermi arc in WTe$_2$. The Weyl points with opposite chiralities are labeled as + (blue) and − (purple). The green arrow represents the spins. **b–d** The voltage hysteresis loop as a function of the in-plane magnetic field under a different DC current ($T = 2$ K). The DC currents in different panels are **b**: $I_{dc} = 10$ μA; **c**: $I_{dc} = 50$ μA; and **d**: $I_{dc} = −50$ μA, respectively. The directions of the magnetic field and current are indicated in the figures. The high- and low-resistance states are also illustrated by the relative orientation of the accumulated spin magnetic moment and moment of the ferromagnetic layer. **e** The voltage difference between the high- and low-resistance states as a function of the DC current. **f** The temperature dependence of the voltage difference between high- and low-resistance states ($I_{dc} = 50$ μA). The error bars in **e** and **f** represent the noise of the field dependence of the voltage hysteresis loop. The thickness of the WTe$_2$ in Device 2 is 23.0 nm. The error bars in **e** and **f** come from the noise in measured voltage

magnetic moment(**s**) and momentum (**k**) are approximately orthogonal to each other. The spin accumulation of the WTe$_2$ conventional surface states is quite similar to that of topological insulators[14–17,41].

According to the spin–momentum correlation presented in Fig. 2c, d, i.e., spin is perpendicular to momentum for both **I**//a-axis and **I**//b-axis, a DC current in WTe$_2$ ribbon devices will accumulate the spins in the surface states. In the WTe$_2$/Al$_2$O$_3$/Fe tunnel junction, a constant DC current (>8 μA) along direction a (or b) between two non-magnetic electrodes (Ti/Au) will lead to a spin accumulation along direction b(or a) in the WTe$_2$ ribbon. Therefore, we can measure the field-dependent resistance of the junction by applying a magnetic field (**H**) perpendicularly to the DC current and applying a much lower AC current (1 μA), as noted in Fig. 2e. Because the resistance of the tunnel junction is governed by the relative orientation of accumulated spin magnetic moments in surface states (**s**) and the moment of the ferromagnetic electrode (**m**)[14–17,41], we could observe the switching of the tunnel resistance of Device 1 (WTe$_2$ 23 nm) at 2 K, as illustrated in Fig. 2f (**I$_{dc}$**//a and **H**//b) and Fig. 2g (**I$_{dc}$**//b and **H**//a). Although the resistance-switching behavior of the b-axis WTe$_2$ ribbon (Fig. 2g) is not as evident as that of the a-axis ribbon (Fig. 2f) of $I_{dc} = 10$ μA, the resistance-switching behavior of the former became immediately clear when the DC current that was applied thereto was increased to $I_{dc} = 30$ μA (Supplementary Fig. 7). To confirm that the hysteretic behavior in magnetic-field-dependent resistance is a direct consequence of the moment switching of the ferromagnetic Fe layer in the tunnel junctions, we measured the magnetic-field-dependent anisotropic magneto-resistance to determine the coercive field of magnetic

layer and the tunneling resistance of WTe$_2$/Al$_2$O$_3$/Fe. Based on the data (Supplementary Fig. 8), we confirmed that the resistance measured via the application of a lower AC current at low temperature is tunneling resistance and the coercive field in the anisotropic MR is approximately 200 Oe, the same as that observed in the resistance switching (Fig. 2f, g, and Supplementary Fig. 8). Upon sweeping the magnetic field, the voltage switching loops detected by applying a weak AC current is a direct consequence of switching magnetic moment from parallel (antiparallel) to antiparallel (parallel) configurations between the magnetic moment of the ferromagnetic electrode and the spin magnetic moment accumulated by conventional surface state.

We also performed the measurement with **I$_{dc}$**//**H**//a and **I$_{dc}$**//**H**//b on Device 1. As depicted in Fig. 2h, no resistance hysteresis behavior was observed for **I$_{dc}$**//**H**//a, further indicating the orthogonal relationship between the accumulated spins of the surface states and the momentum in WTe$_2$. Unexpectedly, we observed the resistance hysteresis loop for **I$_{dc}$**//**H**//b in Device 1 (Fig. 2i). To understand this interesting observation, we turn to the topological Fermi arc in WTe$_2$, a characteristic of the topological Weyl/Dirac semimetals. It has been predicted[14] and demonstrated experimentally[28] that the topological Fermi arcs of WTe$_2$ are along the **Y**-direction of its (001) planes. Based on the calculation (Supplementary Fig. 6), the spin angular momentum is tangential to the linear momentum of topological Fermi arc, as demonstrated in Fig. 3a. Due to the spin–momentum locking effect of topological Fermi arc shown in Fig. 3a, the direction of the accumulated spin should mainly aligned along the b-axis for **I**//b-axis. Therefore, we should observe a resistance loop behavior due to the parallel (or antiparallel) configuration between the

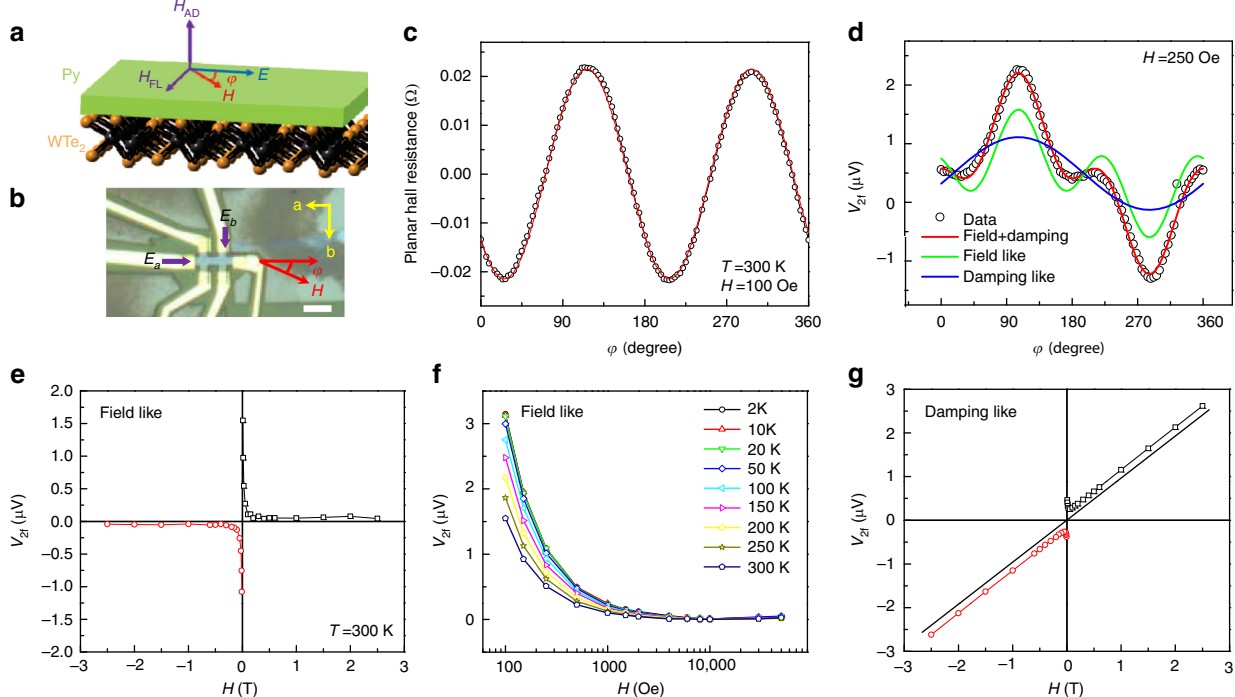

**Fig. 4** Second-harmonic Hall measurement of spin–orbit torques in WTe$_2$/Py. **a** A schematic of the spin orbit torque in WTe$_2$/Py devices under an electric field (**E**) with an in-plane magnetization (**M**). The effective fields of the FL torque and DL torque are also presented. **b** An optical image of the Hall bar device for the second-harmonic Hall voltage ($V_{2f}$) measurement. The AC electrical fields along the various in-plane crystalline axes are indicated. The white scale bar is 10 μm. **c** The typical planar Hall resistance as a function of the azimuthal angle of the magnetic field. **d** The typical angular-dependent $V_{2f}$ ($T$ = 300 K, $H$ = 250 Oe) and the fitted data for both the FL and DL torques. **e** The typical extracted symmetric second-harmonic voltage produced by the FL torque as a function of the magnetic field. **f** The dependence of $V_{2f}$ extracted for the FL torque on the applied magnetic field and temperature. **g** The dependence of $V_{2f}$ extracted for the DL torque on the applied magnetic field

moment of ferromagnetic layer and the accumulated spin magnetic moment, which is generated by topological Fermi arc for $\mathbf{I}//b$-axis.

To confirm the results obtained with $\mathbf{I}_{dc}//\mathbf{H}//b$ for Device 1, we performed the same measurements on Device 2 with different DC currents. In the following, we analyze in greater detail the data obtained from Device 2 (WTe$_2$: 23.0 nm) with an $\mathbf{I}_{dc}//\mathbf{H}//b$ configuration. Because the magnitude of the spin accumulation of the topological Fermi arcs is expected to increase with the density of the DC current, the voltage difference ($\Delta V$) between high- and low-resistance states should increase alongside the increasing $I_{dc}$. Figure 3b, c illustrates the resistance hysteresis loops under DC currents of 10 μA and 50 μA, indicating an amplification of voltage difference as long as increasing of DC current. By reversing the direction of applied DC current bias (to −50 μA), we observed a reversed hysteresis in resistance (Fig. 3d), indicating that the direction of the accumulated spins by Fermi arc was reversed as a result of the spin–momentum locking effect. The inset in Fig. 3b–d depicts the relative orientation between accumulated spin magnetic moment in WTe$_2$ and the moment of ferromagnetic layer. The dependence of $\Delta V$ on the DC current obtained at 2 K is plotted in Fig. 3e. As we expected, a nearly linear dependence is observed in a wide range of currents (Supplementary Fig. 9). According to the theoretical calculation, the Weyl points and Fermi arc in WTe$_2$ gradually vanish as the temperature increases[30]. We found that the voltage difference ($I_{dc}$ = 50 μA) decreased with increasing temperature and eventually vanished at temperatures above 15 K (Fig. 3f and Supplementary Fig. 10). The reduction of $\Delta V$ should be ascribed to the suppression of 2D electron transport($T < 50$ K) and topological Fermi arc($T < 100$ K) states on the top surface of WTe$_2$ at higher temperatures[30]. To consolidate our conclusion,

we repeated the experiment on Device 3 (WTe$_2$: 17 nm, Supplementary Fig. 11) with the same observation (as indicated in Fig. 2). These results further confirm the spin–momentum locking and spin accumulation in the topological Fermi arc of WTe$_2$.

**Anisotropic spin–orbit torques in WTe$_2$/Py**. To further explore the exotic properties of the surface states of WTe$_2$ and their potential applications, we fabricated WTe$_2$/Py Hall bar devices (Supplementary Fig. 5) and investigated the spin–orbit torque in WTe$_2$/Py devices. Due to the fact that the Fermi arcs are along the **Y**-direction of the (001) planes of WTe$_2$ and that the spin momentum is tangential to the topological Fermi arcs (Fig. 3a)[23,28]. Both the spin–momentum locking and spin–orbit torques induced by the open Fermi arcs must be highly anisotropic and significantly different from the isotropic topological surface states in the topological insulators.

As is well known[3], two types of spin–orbit torques, namely FL torque ($\boldsymbol{\tau}_F = \mathbf{m} \times \boldsymbol{\sigma}$) and DL torque ($\boldsymbol{\tau}_D = \mathbf{m} \times (\mathbf{m} \times \boldsymbol{\sigma})$) are often observed in ferromagnet/heavy-metal devices. We therefore expected to observe both torques in the WTe$_2$/Py devices. To examine the quality of the WTe$_2$/Py devices, we imaged the cross-section (Supplementary Fig. 2) and measured the basic physical properties (Supplementary Fig. 12) of the devices. Figure 4a schematically presents the directions of the torques, magnetization, and applied electric field in our measurement configuration. Figure 4b contains the optical image of the devices used in this study. The first-harmonic Hall resistance, or planar Hall resistance, measured on the WTe$_2$/Py Hall bar device (Fig. 4b), closely follows the $\sin 2\varphi$ dependence (Fig. 4c), suggesting that the moment of Py is always aligned with the external magnetic field for $H > 100$ Oe. The second-harmonic Hall voltage $V_{2f}$, as a

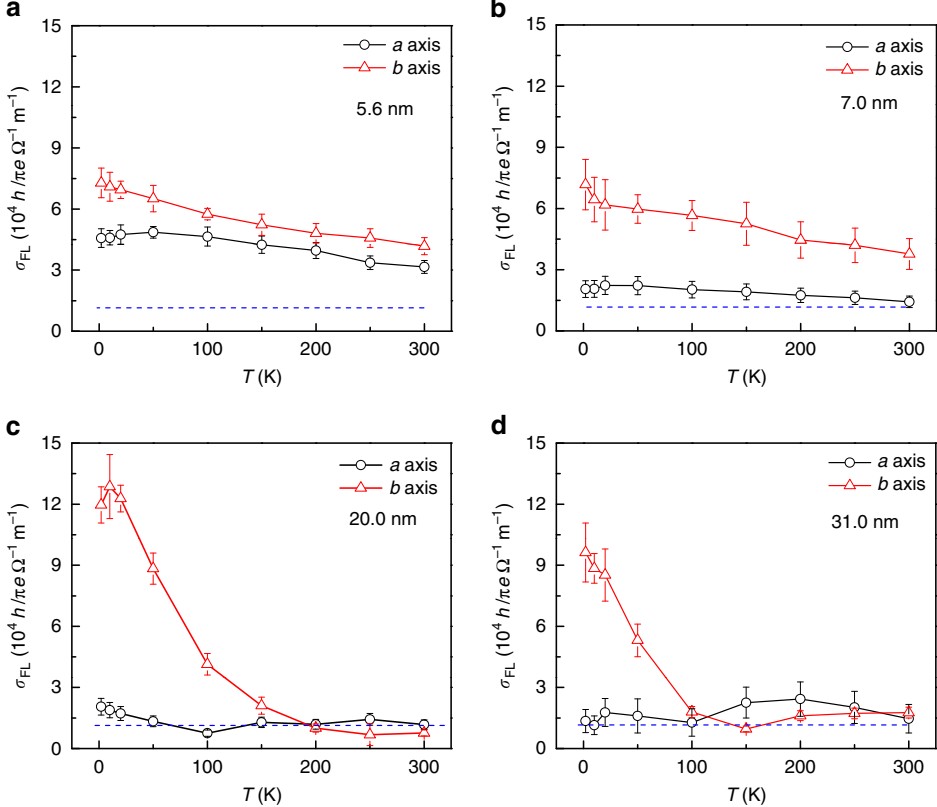

**Fig. 5** Anisotropic spin conductivity in WTe$_2$/Py devices. The spin conductivity $\sigma_{FL}$ calculated using the extracted FL torque in WTe$_2$/Py with the current along the $a$- and $b$-axes. The thicknesses of the WTe$_2$ in the Hall devices are 5.6 nm (**a**), 7.0 nm (**b**), 20.0 nm (**c**), and 31.0 nm (**d**), respectively. The thickness of the Py in the devices is 6 nm. The dashed blue line represents the spin conductivity calculated using the estimated Oersted field. The error bar represents the combination of the 20% standard error and the fitting process

function of the in-plane azimuthal angle $\varphi$, can be accommodated by accounting for the two torque components (Supplementary Note 1)[43,44].

$$V_{2f} = \left[ \frac{-H_{FL}\cos(\theta+\theta_0)}{H - H_A} R_P \cos 2(\theta+\theta_0) + \frac{1}{2} \frac{H_{DL}\cos(\theta+\theta_0)}{H_K - H} R_A \right] I$$
$$= \left[ V_{FL}\cos(\theta+\theta_0)\cos 2(\theta+\theta_0) + \frac{1}{2} V_{DL}\cos(\theta+\theta_0) \right] I . \quad (1)$$

By fitting the data in Fig. 4d to Eq. 1, we obtained the amplitude of $V_{FL}$ and $V_{DL}$, where $V_{FL}$ and $V_{DL}$ are the voltages that originated from the FL and DL torques, respectively (Supplementary Fig. 13). The fitting curves are also plotted in Fig. 4d. Clearly, the fitted FL term $V_{FL}$ (Fig. 4e) is quite centrosymmetric and rapidly decreases as the magnetic field increases (Supplementary Fig. 13), as one would expect based on Eq. 1. It is, therefore, straightforward to obtain the values of the FL torque at different temperatures from the data in Fig. 4f by using Eq. 1. From the extracted field dependence of $V_{DL}$ (Fig. 4g), the step-like shift across the zero magnetic field corresponds to the Joule heating via the anomalous Nernst effect (ANE)[43]. More importantly, a very strong linear background is superimposed onto the ANE component. This strong linear background prevents us from extracting reliable DL torques[45]. We believe that this strong linear background in the field-dependent $V_{DL}$ is due to the chiral-anomaly-induced giant planar Hall resistivity in WTe$_2$ (Supplementary Fig. 14). The giant planar Hall resistivity induced by chiral anomaly in topological semimetals has been predicted and confirmed by recent experiments[46–48].

The results associated with FL spin torque may now be considered. In Fig. 5, we plot the temperature dependence of the

spin conductivity associated with the FL spin–orbit torque ($\sigma_{FL} = H_{FL} \cdot M_S \cdot t/E$) with the current along the $a$- and $b$-axes of WTe$_2$ in the devices of WTe$_2(t)$ /Py(6 nm) with $t = 5.6$ nm, 7.0 nm, 20.0 nm, and 31.0 nm. The most prominent feature in Fig. 5 is that for all four samples, spin conductivity is strongly anisotropic at low temperatures, particularly for the devices with thick WTe$_2$. The weak and nearly temperature-independent $\sigma_{FL}$ is observed in all four devices when the current runs along the $a$-axis of the WTe$_2$. When the current is applied to the $b$-axis of the WTe$_2$, the values of $\sigma_{FL}$ are weakly temperature-dependent in all devices. The most surprising and interesting features were observed in the data obtained from the two devices with thick WTe$_2$ (20.0 nm and 31.0 nm). Not only much larger but also strongly temperature-dependent $\sigma_{FL}$ was observed at low temperatures (Supplementary Fig. 15). Since the saturation magnetization of Py layer, below 50 K, is weakly dependent on temperature, the enhanced spin conductivity with the current along the $b$-axis must be attributed to the increase in FL torques. The obtained spin conductivity at low temperatures is less than that in the topological insulator Bi$_2$Se$_3$ but nearly one order of magnitude higher than that in the MoS$_2$ and WSe$_2$ monolayers[49,50].

To understand the enhancement of the spin–orbit torque at low temperatures, we have to exclude other extrinsic factors such as the possible spin–orbit torques caused by the capping layer Ru. The contribution of the Ru layer to the measured FL spin conductivity is negligibly small (Supplementary Fig. 14). In some cases, the Oersted field generated by the current flowing in the non-magnetic layers may also contribute to the measured FL torques[45,51]. We thus estimated the Oersted field using Ampere's law ($\tau_{Oe} = \mu_0 I/2A$)[45,51], where $I$ is the current flowing in the

non-magnetic layer and A is the width of the Hall device. In our experiments, the width of $WTe_2$ was approximately 5 μm, and the total AC current was about 0.5 mA, which led to an Oersted field of 0.62 Oe by assuming that the current flows only in $WTe_2$. If we consider that only a fraction of the current flowing in the $WTe_2$ layer determined by the resistivity of $WTe_2$ and Py (Py: ~80 μΩ cm; a-axis $WTe_2$: ~148 μΩ cm; b-axis $WTe_2$: ~121 μΩ cm, Supplementary Fig. 16), the resulted Oersted field must be smaller than 0.62 Oe (Supplementary Fig. 17). The calculated contribution of the Oersted field to the spin conductivity (dashed line) is also plotted in Fig. 5 for a direct comparison with the spin conductivity associated with FL torques. It is evident that although the contribution of the Oersted field is comparable to that obtained for I//a, the much larger and strongly temperature-dependent $\sigma_{FL}$ for I//b and T<100 K must be caused by the intrinsic properties of $WTe_2$.

Based on our previous work on $WTe_2$, we know that if the thickness of $WTe_2$ is in the range of approximately 10 nm to approximately 40 nm, we will observe the Weyl semimetal feature in the low-temperature magneto-transport properties[28], including the existence of Fermi arc only along the b-axis and the spin–momentum locking in the Fermi arcs (Fig. 3a). The 2D nature of the electronic transport (Fig. 1) in $WTe_2$/Py indicates that the topological Fermi arc remains active at the interface of $WTe_2$/Py, which leads to an additional spin accumulation at the interface of the I//b-axis. Therefore, a dramatic enhancement of spin–orbit torques should be expected at low temperatures and I//b-axis, as observed in Fig. 5. To further demonstrate that the topological Fermi arcs indeed enhance the spin–orbit torques, we fabricated Hall bars in which the current can only flow along a direction deviating by approximately 31° (Hall bar) from the b-axis. Interestingly, to best fit the data using Eq. 1 to the angular-dependent second-harmonic Hall voltage obtained at T < 100 K, we had to shift the fitted phase angle $\theta_0$ of high temperature curves (T > 100 K, $\theta_0$: 68.3°) to low temperature ones (T < 100 K, $\theta_0$: 31.8°) approximately by 36°(Supplementary Fig. 18 and Note 1). The phase-shift of approximately 36° is apparently close to the 31°, the angle between the current and b-axis of $WTe_2$ in the devices. Since the topological Fermi arcs exist only along the b-axis of $WTe_2$ and at temperatures below 100 K[30], we can conclude that the topological Fermi arcs (along the b-axis) significantly contributed to the enhanced spin–orbit torques at T<100 K based on our observation presented above.

## Discussion

In addition to the key feature in Fig. 5, i.e., the enhancement of spin conductivity (I//b) at low temperatures, another interesting phenomenon should be noticed. The relatively large spin conductivity and FL torques in thinner $WTe_2$ devices is observed at high temperature for both I//a-axis and I//b-axis configurations. In particular, the $\sigma_{FL}$ obtained at high temperatures in devices of thinner $WTe_2$ (t = 5.6 nm and 7.0 nm) with I//a-axis is larger than that in the thicker $WTe_2$ devices (t = 20.0 nm and 31.0 nm).

Due to the strong thickness dependence of the resistivity of $WTe_2$, i.e., the resistivity of $WTe_2$ greatly increases when decreasing the thickness of $WTe_2$[28,52], the fraction of current that flows in $WTe_2$ in the Hall bar devices will therefore significantly decrease with decreasing thickness of $WTe_2$, leading to an even smaller spin conductivity. As we discussed above, for the devices with a $WTe_2$ thickness of less than 10 nm, the topological surface state should vanish, which again will reduce the $\sigma_{FL}$. Therefore, the large $\sigma_{FL}$ observed in devices with thin $WTe_2$ cannot be the ascribed to the spin–momentum locking and spin accumulation by Fermi arcs.

Previous studies have also suggested that the Rashba effect at the interface is one of the origins of the spin–orbit torques in the

monolayer $MoS_2$ ($WSe_2$)/Py bilayers[50]. The low-temperatures data in Fig. 1 indicate the existence of 2D transport in our devices, which supports the existence of a 2D Rashba interface[53,54]. With decreasing the thickness of $WTe_2$, the electrical conductivity in thinner $WTe_2$, decreases accordingly. Consequently, the fraction of the current flowing in the highly conductive Rashba interface (Fig. 1 and Supplementary Fig. 4) increases[52]. Therefore, the dominance of the 2D Rashba interface in thinner $WTe_2$ devices should be the key origin of the stronger FL torques in thinner $WTe_2$ devices.

One may argue that the highly anisotropic Rashba effect in $WTe_2$/Py, induced by the highly anisotropic crystal structure ($WTe_2$)[55], is also the dominant mechanism that enhances the spin–orbit torques at low temperatures when I//b-axis. However, we should also note that the temperature-dependence of FL torques is quite weak with the current along the a-axis but strong along the b-axis, which is inconsistent with the temperature-independent feature of the Rashba-interface-dominated spin–orbit torques[50]. One may also argue that the spin Hall effect of $WTe_2$ can also account for the enhancement of spin–orbit torques at low temperatures. However, the spin Hall effect usually contributes to the DL torques rather than to FL torques[56]. The results shown in Fig. 5 clearly point to FL torques, which excludes the spin Hall effect as the origin of the enhancement of spin conductivity at low temperatures (b-axis devices). After excluding the contributions of the anisotropic Rashba effect at the interface and the spin Hall effect of $WTe_2$, we can ascribe the greatly enhanced spin conductivity at low temperatures in thicker $WTe_2$ devices with the current along the b-axis to the Fermi-arc-assisted spin–orbit torques. The weak discrepancy of critical temperatures for Fermi arc state in spin momentum locking (Fig. 3f: ~15 K) and spin orbit torques (Fig. 5c, d: ~100 K) should be ascribed to different interface of devices (detailed discussion in Supplementary note 2 and Supplementary Figs.19-20 for more devices with different $WTe_2$ thickness).

It has been shown that several factors could affect the transport of the Fermi arc states. For example, the scattering between the Fermi arc surface states and bulk states for a Weyl semimetal would lead to dissipation in the transport[57]. In addition, such scattering has sensitive dependence on the arc geometry[58]. It has been found that straight arc geometry is very disorder tolerant. For $WTe_2$ studied here, its Fermi arc states coexist with the bulk states due to its type-II nature (Supplementary note 3 for the conduction ratio of Fermi arc to bulk states), and the arc shape is not quite straight. Hence, we expect that the scattering effects would be important in the dissipative transport. However, those previous theories are about type-I Weyl semimetals. A theory on type-II Weyl materials would be desired to give more appropriate account of our experiment.

To conclude, we have obtained the greatly enhanced spin–orbit torques at low temperature in $WTe_2$/Py devices from magnetic transport measurements when the current is flowing along the b-axis of $WTe_2$. The greatly enhanced spin conductivity can be interpreted by the effect of spin–momentum locking in the topological Fermi arcs states of the $WTe_2$ Weyl semimetal at the interface. This study should greatly contribute to research on new materials with high spin-charge conversion efficiency for magnetization reversal. Our work is directed toward the potential application of Weyl physics in Spintronics.

## Methods

**Device fabrication**. $WTe_2$/$Al_2O_3$/Fe tunnel junction: $WTe_2$ single crystals grown with the vapor chemical transport method were obtained from HQ Graphene Company. After exfoliating the $WTe_2$ flakes using $SiO_2$ (285 nm)/Si, 3-nm-thick Al was coated onto the $WTe_2$ flakes by e-beam evaporation and automatically

oxidized into $Al_2O_3$. The in-plane crystal orientation (*a*- and *b*-axes) was determined using the angle-dependent polarized microscopic Raman spectrum (Hariba LABRAM HR spectrometer). To realize the DC current flowing along the $WTe_2$ ribbons with different crystal orientations (*a* or *b*), $WTe_2$ ribbons with widths of approximately 1 μm and lengths of approximately 12 μm were patterned using standard electron-beam lithography (EBL; Crestec-9000), which was followed by the process of etching with Ar gas to remove the excess $WTe_2$ flakes. The non-magnetic electrodes were patterned using a second EBL process, which was followed by Ti (10 nm)/Au (70 nm) e-beam evaporation. Before the deposition of Ti/Au in the second step, we etched the $Al_2O_3$ thin layer on top of $WTe_2$ to ensure the good contact between electrodes and $WTe_2$. Next, we performed the third EBL and coated 20-nm-thick $SiO_2$ on the edges of the $WTe_2$ ribbons in order to insulate the top ferromagnetic electrodes and the edge of the $WTe_2$. Next, the ferromagnetic electrode was written using a fourth EBL process, followed by the deposition of magnetic electrodes Fe (6 nm)/Ti (4 nm) using e-beam evaporation. The final device is presented in Supplementary Fig. 5.

$WTe_2$/Py bilayers: First, the multilayered $WTe_2$ flakes were exfoliated onto $SiO_2$ (285 nm)/Si substrates. To minimize the oxidation of $WTe_2$, we coated Py (6 nm)/Ru (4 nm) onto the $WTe_2$ flakes by off-axis sputtering immediately after the exfoliation. This 4-nm-thick Ru was used to protect the Py from oxidation in air. The in-plane crystal orientation (*a*- and *b*-axes) was determined using the angle-dependent polarized microscopic Raman spectrum, following the same process employed for the $WTe_2$/$Al_2O_3$/Fe devices. To extract the spin–orbit torques along different crystal orientations, we fabricated Hall bars with widths of 5 μm using standard EBL patterning and Ar etching. Then, the electrode was patterned using the second EBL process, after which we deposited the Ti (10 nm)/Au (70 nm) electrodes using e-beam evaporation. The device fabrication process is schematically illustrated in Supplementary Fig. 5.

**Measurements**. The thickness of the $WTe_2$ flakes was determined by using an atomic force microscope. The cross-sections of the $WTe_2$/Py bilayers were imaged using a monochromated, Cs-corrected high-resolution scanning transmission electron microscope (Titan 80-300, FEI). The magnetotransport properties, including the planar Hall effect, anomalous Hall effect, MR, and second-harmonic planar Hall voltage, were measured using a Quantum Design Physical Property Measurement System (PPMS-Dynacool) in a temperature range of 2–300 K and a magnetic field range of 0–14 T. The magnetic-field-dependent spin voltage between the ferromagnetic layer and nonmagnetic electrode in $WTe_2$/$Al_2O_3$/Fe was measured under an AC current of 1 μA. The DC current varied from 8 μA to 50 μA. The second-harmonic Hall measurements of spin–orbit torques were measured by lock-in SR 830 (Stanford) under an AC frequency of 87.8 Hz and an AC current of about 0.5 mA. The angular dependence of the second-harmonic Hall voltage and MR were measured with a sample rotator.

**First-principle calculations**. The correlation between the spin and momentum of the surface states in $WTe_2$ was computed by first-principle calculation to confirm its spin–momentum locking effect. The first-principles calculations are based on the density functional theory and use the projector augmented-wave method[59] as implemented in the Vienna Ab Initio Simulation Package[60,61]. The generalized gradient approximation with the Perdew–Burke–Ernzerhof realization[62] was adopted for the exchange-correlation potential. The plane-wave cutoff energy was set to 450 eV. A Monkhorst-Pack k-point mesh[63] of size $15 \times 8 \times 4$ was used for Brillouin zone sampling. The energy convergence criterion was set to $10^{-5}$ eV. The crystal structures were optimized until the forces of the ions were less than 0.01 eV/Å. For the first-principles calculations, we took the experimental lattice parameters ($a = 3.477$ Å, $b = 6.249$ Å, and $c = 14.018$ Å) of $WTe_2$[64]. The surface states and spin–momentum locking effect were investigated by constructing the maximally localized Wannier functions[59,65,66] using the WANNIER-TOOLS package[67] combined with an iterative Green's function method[68,69]. To study the electronic properties of the interface of the $WTe_2$/Py (or Au) heterostructure, we used a $WTe_2$/Au (110) heterostructure slab model containing four $WTe_2$ layers ($4 \times 1$ supercell) and 10 Au atom layers ($3 \times 2$ supercell). A vacuum layer larger than 15 Å was used to eliminate the interaction between adjacent images, and the atoms were fixed—except for those on a few layers near the interface. The cutoff energy for the plane-wave basis set was set to 250 eV for the heterojunction system, and the Brillouin-zone was sampled using a Monkhorst-Pack k-point mesh[63] of $2 \times 2 \times 1$. The lattice constant exerting the least amount of stress on the cell was selected. The forces exerted on the atoms were less than $10^{-2}$ eV/Å, and the energy convergence criterion was set to $10^{-5}$ eV.

## Data availability
The authors declare that the data supporting the findings of this study are available within the paper and its Supplementary Information files.

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

## Acknowledgements

We would like to thank Prof. Zhixun Shen and Prof. Arun Bansil for their useful discussions. We thank Y.L. Zhao, J.L. Zhang, and Q. Zhang for the useful discussions and some experimental support. The work reported was funded by King Abdullah University of Science and Technology (KAUST), Office of Sponsored Research (OSR) under the Award Nos. CRF-2015-SENSORS-2709 (KAUST) and CRF-2015-2626-RG4.

## Author contributions

P.L. conceived the idea, designed the experiments, and X.X.Z. supervised the project. P.L. and X.X.Z. wrote the manuscript. P.L. and C.H.Z. performed the sample fabrication and transport measurements. P.L. and Y.W. made the second harmonic Hall measurements. W.K.W., Z.M.Y., and S.A.Y. made the calculation. J.W.Z. imaged the cross-sectional TEM. S.F.Z. made the simulation of Oersted field. P.L. and Y.W. performed the analysis on the data. P.L., X.X.Z., and all other authors contributed to the discussion of the results and improvement of the manuscript.

## Additional information

**Competing interests:** The authors declare no competing interests.

