## [Peer Review File · Nature Communications]

Editorial Note: Several images have been redacted as indicated to protect copyright claims. The images can be viewed in the publications in which they originally appear, as indicated at the place of redaction.

Reviewers' comments:

Reviewer #1 (Remarks to the Author):

Li et al reported the study of spin-orbit torque at the interface between WTe₂/Py, and attributed the large field torque to the spin-momentum locking properties. These results are potentially interesting for the field of topological spintronics. However, there are several major issues and inconsistencies in this manuscript.

1) Critical experimental data are lacking for the conclusion in this manuscript. The authors claimed that the large field torque to the spin-momentum locking properties. To support this conclusion, the samples of the same or similar thicknesses have to be investigated for spin-momentum locking and spin-orbit torque.

For example, in Fig. 5, for thinner thicknesses (5.6 nm and 7.0 nm), a smaller anisotropy is observed for the whole temperature region. While for thicker samples (20 nm and 31 nm), the anisotropy is quite significant only at low temperatures ($T < \sim 100$ K). If one wants to claim this anisotropy arises from the spin-momentum locking, one has to do the spin-momentum locking (as in Fig. 2 and 3) on the same or similar thicknesses (from ~ 5 nm to ~ 30 nm), and then to compare these two measurements.

Without these experimental data, the conclusion is not well supported.

2) The critical temperatures between Fig. 3f and Fig. 5c are not consistent for the similar thicknesses (23 nm vs. 20 nm). For the spin-momentum locking measurements, the signal is observed below 20 K, indicating the anisotropy below 20 K. However, in Fig. 5c, the strong anisotropy is observed below 150 K, and there is a slightly downward turn from ~ 10 K to ~ 2 K.

These are certainly inconsistent with each other.

3) Why there are anisotropy for 5.6 nm and 7.0 nm? For these thinner layer, they are not expected to be Weyl semimetals (ref. 27).

4) For these materials, there are bulk conducting states. Have the authors estimated the contributions from bulk conducting states?

What are the carrier densities for surface and bulk states?

5) The technique to probe the spin-momentum locking was first used in Nat Nanotechnol 9, 218-224 (2014), but it was criticized by several later reports, including fringe field effect (E. K. de Vries, et al, Phys. Rev. B, 92, 201102 (2015)), and other spurious effects (Pengke Li and Ian Appelbaum, Phys. Rev. B 93, 220404(R) (2016)).

Have the authors ruled out those possibilities?

Reviewer #2 (Remarks to the Author):

It is an interesting work and addresses a key topic. The field of Weyl semimetals has recently experienced a surge in all sorts of theoretical and experimental activities since the original proposal on the existence of Weyl points and Fermi arcs states in iridate materials. However, most work up to date has been done to detect Fermi arcs themselves and study chiral anomaly effect, two most striking features of Weyl semimetals, but the topological nature of the Fermi arcs states suggests their genuine applications in surface transport phenomena (as is the case of the topological insulators), something that is only beginning to appear. Therefore the present manuscript is important, timely, and potentially interesting to a broad physics community. However, the authors need to pay attention to the following points before the decision on publication is made: First, spin -momentum locking in topological insulators follows directly from the Hamiltonian of the surface state written in the form $H = v [\mathbf{k} \cdot \mathbf{e}_z] \sigma_y$. There is no such Hamiltonian for the arc states, hence there is no spin momentum locking per se. On general ground, one can only expect that the spins along the arc should wind from one Weyl point to another Weyl point since the arc connect the points of opposite chirality. This has to be clearly explained in the manuscript and calling this effect "locking" is confusing. Second, the spin accumulation in topological insulators was studied theoretically in great detail in a recent Nature Communications 7, 10878 (2016) where the transfer of spin through the bulk has been emphasized as the primary mechanism of the spin accumulation. The authors here assume that it is a single surface phenomenon but bulk boundary correspondence, the genuine feature of topological systems seems to be missing in such assumption. The authors should discuss this point in greater details. Third, surface transport properties have been studied recently in a series of works [Phys. Rev. B 93, 235127 (2016), Phys. Rev. B 97, 085142 (2018)]. As the authors study a 2D transport in their system, and highlight rather high values for spin conductivity, it would be good to see a discussion in relation to the theoretical insights brought by these recent studies.

Reply to the comments of NCOMMS-18-15480-T

Response to Reviewer #1 :

Reviewer #1: Li et al reported the study of spin-orbit torque at the interface between WTe₂/Py, and attributed the large field torque to the spin-momentum locking properties. These results are potentially interesting for the field of topological spintronics. However, there are several major issues and inconsistencies in this manuscript.

Comment 1: *1) Critical experimental data are lacking for the conclusion in this manuscript. The authors claimed that the large field torque to the spin-momentum locking properties. To support this conclusion, the samples of the same or similar thicknesses have to be investigated for spin-momentum locking and spin-orbit torque.*

For example, in Fig. 5, for thinner thicknesses (5.6 nm and 7.0 nm), a smaller anisotropy is observed for the whole temperature region. While for thicker samples (20 nm and 31 nm), the anisotropy is quite significant only at low temperatures ($T < \sim 100$ K). If one wants to claim this anisotropy arises from the spin-momentum locking, one has to do the spin-momentum locking (as in Fig. 2 and 3) on the same or similar thicknesses (from ~ 5 nm to ~ 30 nm), and then to compare these two measurements.

Without these experimental data, the conclusion is not well supported.

Reply: Thank you for suggestion to make our work improved. Following your suggestion, we did more experiments to examine the spin momentum locking effect in WTe₂ devices with the similar thickness as that in spin orbit torques. The thicknesses of our newly measured samples with resistance hysteretic behaviors are 10.0 nm, and 28.0 nm respectively. We can't observe

the resistance hysteresis feature in the samples with thickness thinner than 7.0 nm, although we measured several thin WTe_2 devices with thickness ranging from 4.0 to 7.0 nm.

Figures R1-R3 gives the electrical detection of spin momentum locking in 10.0 nm thick WTe_2 device. Figure R1 shows the electrical detection of surface states with different measurement configurations, in which a DC current of $20 \mu\text{A}$ was applied. Except for the case of I_{dc}/a and $H//a$ (Fig. R1 (d), we indeed observed the resistance hysteretic feature for the rest experimental configurations. It is clear that all the results obtained with different configurations in Figure R1 display the same characteristics as that presented in Figures 2f to 2i in the main text.

Figure R1, Electrical detection of spin-momentum locking in $\text{WTe}_2(t=10.0 \text{ nm})/\text{Al}_2\text{O}_3/\text{Fe}$ tunnel junctions with different measurement configuration. (a) I_{dc}/b , $H//a$; (b) I_{dc}/b , $H//b$; (c) I_{dc}/a , $H//b$; (d) I_{dc}/a , $H//a$. All the measurements are performed at 2 K and under a DC current of $20 \mu\text{A}$.

To further demonstrate the spin-momentum locking effect in topological Fermi arc, we varied the strength of dc currents applied to the same sample ($t=10.0$ nm) with both dc current (I_{dc}) and magnetic field (H) along b -axis, as shown in Figure R2a. Figure R2b summarizes the dependence of the voltage difference between the high- and low-resistance states on the DC current. The nearly linear dependence of voltage difference on DC currents is in good agreement with that in Figure 3e in main text. The relative large error bars in 10.0 nm thick sample should be related closely to the fact that the thickness of 10 nm is near the critical thickness that Weyl semimetal states and topological Fermi arcs are gradually vanishing.

Figure R2 Electrical detection of spin-momentum locking in $\text{WTe}_2(t=10.0 \text{ nm})/\text{Al}_2\text{O}_3/\text{Fe}$ tunnel junctions with different DC currents at 2 K. During the measurements, an AC current of 1 μA is applied to measure the tunnel resistance. (a) The voltage as a function of in-plane magnetic field with different DC currents. (b) The linear dependence of voltage difference on the DC current.

Figure R3a shows all the magnetic field dependent voltage at different temperatures obtained on the same 10.0 nm thick sample with a DC current of 20 μA . Figure R3b summarizes

the voltage difference as a function of temperature. The critical temperature that hysteresis behavior vanishes is as low as 7-10 K in 10.0 nm thick WTe_2 sample.

Figure R3 Electrical detection of spin-momentum locking in $WTe_2(t=10.0 \text{ nm})/Al_2O_3/Fe$ tunnel junctions at different temperatures. During the measurements, a DC current of $20 \mu A$ and an AC current of $1 \mu A$ are applied to measure the tunnel resistance. (a) The voltage as a function of in-plane magnetic field at different temperatures. (b) The temperature dependent voltage difference. The critical temperature that hysteresis behavior vanishes is near 7-10 K.

Figures R4-R5 summarize the similar data in 28.0 nm thick sample as that in 10.0 nm (Figure R1-3) and 23.0 nm thick samples (Figure 2-3 in main text). Obviously, under the application of a DC current of $20 \mu A$, the hysteresis loop behavior has been observed clearly for three measurement configurations ($I_{dc} // a, H // b$; $I_{dc} // b, H // a$; $I_{dc} // b, H // b$), as shown in Figure R4. And the results presented in Figure R5 show that the critical temperature that hysteresis behavior ($I_{dc}=20 \mu A$) vanishes increased near 15-20 K, compared to the critical temperature of 7-10 K in 10.0 nm WTe_2 (Figure R3) and 12-15 K in 23.0 nm WTe_2 (Figure 3 in main text).

Figure R4 Electrical detection of spin-momentum locking in $WTe_2(t=28.0 \text{ nm})/Al_2O_3/Fe$ tunnel junctions with different measurement configurations. (a) $I_{dc} // b$, $H // a$; (b) $I_{dc} // b$, $H // b$; (c) $I_{dc} // a$, $H // b$; (d) $I_{dc} // a$, $H // a$. All the measurements are performed at 2 K and under a DC current of $20 \mu A$.

Figure R5 Electrical detection of spin-momentum locking in $WTe_2(t=28.0 \text{ nm})/Al_2O_3/Fe$ tunnel junctions at different temperatures. During the measurements, a DC current of $20 \mu A$ and an AC current of $1 \mu A$ were applied to measure the tunnel resistance. (a) The voltage as a function of in-plane magnetic field at different temperatures. (b) The temperature dependent voltage difference. The critical temperature that hysteresis behavior vanishes is near 15-20 K.

Therefore, the data shown in Figure R1-R5 demonstrates that the reproducibility of electrical detection of spin momentum locking in conventional surface states and topological Fermi arc states in WTe_2 . The critical temperatures that hysteresis behavior vanishes are 7-10 K, 12-15 K and 15-20 K for 10.0 nm, 23.0 nm and 28.0 nm thick samples. The critical temperature does not seem to agree well with the observation in spin-orbit-torques in Figures 4-5 in main text. Nevertheless, the different interfaces in heterojunctions will account for the difference in both spin momentum locking and spin orbit torques. We will discuss it below.

We added the new data into Supplementary Figures 19 and 20.

***Comment 2:** 2) The critical temperatures between Fig. 3f and Fig. 5c are not consistent for the similar thicknesses (23 nm vs. 20 nm). For the spin-momentum locking measurements, the signal is observed below 20 K, indicating the anisotropy below 20 K. However, in Fig. 5c, the strong anisotropy is observed below 150 K, and there is a slightly downward turn from ~10 K to ~2K.*

These are certainly inconsistent with each other.

Reply: Thank you for your comment to improve our manuscript. As we observed and discussed above, the critical temperatures that hysteresis behavior vanishes are 7-10 K, 12-15 K and 15-20 K for 10.0 nm, 23.0 nm and 28.0 nm thick samples. The critical temperature does not seem to agree well with the observation in spin-orbit-torques in Figures 4-5 in main text. There are two reasons that account for the difference in both experiments.

First of all, we have to point out that the interfaces in the samples used to measure two effects are different, i.e. two interfaces in the sample of ferromagnet-insulator-Weyl used in spin momentum locking measurements, and one interface in the sample of ferromagnet-Weyl used in spin-orbit torques measurements. The effective spin polarization detected by ferromagnet and ferromagnet/insulator should be different.¹ Only when the thickness of the tunnel barrier is thick enough, the spin polarization by ferromagnet/insulator will approach the bulk spin polarization of ferromagnet.¹ Normally, the effective spin polarization in ferromagnet/insulator is smaller than that in bulk ferromagnet. Therefore, the critical temperature obtained in spin momentum locking experiment using a device of WTe_2/Al_2O_3 should be smaller than that obtained from the spin orbit torque measurement using a WTe_2/Py device for the thick tunneling barrier lay of Al_2O_3 (3 nm).

Second, during the device fabrication process, there are two more E-beam lithography (EBL) steps in making the devices used in spin momentum locking detection than in making the devices used in spin orbit torques measurements. Particularly, the WTe_2/Al_2O_3 layer has to be exposed to the e-beam lithography photoresist (PMMA). The quality of WTe_2 crystal might be

deteriorated by the fabrication process, which might consequently damage the robust of topological Fermi arc states. Therefore, in comparison the critical temperature observed in the spin orbit torques measurements, the decrease of critical temperature observe in the measurements of spin momentum locking should be expected.

Most importantly, with the increase of WTe₂ thickness, the critical temperature in spin momentum locking detection increases from 7-10 K (10.0 nm) to 15-20 K (28.0 nm). This tendency agrees well with the observation in spin orbit torques measurements (Figure 5 in main text).

As to the downward turn in spin conductivity, we believe that the slight downward turn in the temperature dependent spin conductivity (*b*-axis) from ~10 K to ~ 2K should be ascribed to the experimental error and fitting error (extraction of field like torque). Obviously, the fitting error bar of the data (*b*-axis) at 10 K in Figure 5c is relatively larger than that in other cases.

The reason for small discrepancy of critical temperature in both experiments is added in line 22 page 18 in main text and Supplementary note 2.

***Comment 3:**3) Why there are anisotropy for 5.6 nm and 7.0 nm? For these thinner layer, they are not expected to be Weyl semimetals (ref. 27).*

Reply: Thank you for your comment. According to Ref. 27 and our previous studies,^{2,3} the band crossing will be gapped gradually with the decreasing thickness of WTe₂. We found that the characteristic of the Weyl semimetals, i.e. chiral anomaly, Weyl orbit quantum oscillations, will fade out in the WTe₂ flakes with thickness thinner than 10 nm.

The anisotropy in spin conductivity (σ_b / σ_a) with current along *b*- and *a*- axes in 20.0 nm and 31.0 nm thick samples in the low temperature region are 7.5 and 7.2, respectively, whereas the anisotropy in 5.6 nm and 7.0 nm thick samples are 1.5 and 3.0, respectively. It is clear that

the anisotropy in spin conductivities in thinner samples (5.6 nm and 7.0 nm) is much weaker than that in thicker samples (20.0 nm and 31.0 nm). Most importantly, the ratio of $\sigma_{2K} / \sigma_{300K}$, which is a Hallmark of enhancement in spin conductivity by topological Fermi arc states, are 1.68, 1.85, 16.3 and 7.0 for 5.6, 7.0, 20.0 and 31.0 nm thick *b*-axes samples, respectively. Evidently, the ratio of $\sigma_{2K} / \sigma_{300K}$ in thicker samples is much higher than that in thinner ones.

As we discussed in main text (third paragraph of Discussion), the Rashba effect in thinner samples should be responsible for the relative high spin conductivity at both directions. Although the anisotropies in spin conductivity of thinner samples are smaller than those in thicker samples ($t=20.0$ and 31.0 nm), the weak anisotropies in 5.6 and 7.0 nm samples should be attributed to the difference in current ratio across WTe₂/Py Rashba interface along both *a*- and *b*-axis. The difference in current ratio across Rashba interface should be closely related to the weak difference in temperature dependent resistivity of thin WTe₂ layer, as shown in Figure R6 and Supplementary Figure 16b.

Figure R6 Temperature dependence of resistivity in thin WTe₂ ($t=7.0$ nm) along different directions.

Apparently, we observed the hysteretic behaviors in resistance in 10.0 nm and 28.0 nm thick samples, as shown in Figures R1-R5, and in 23.0 nm sample as shown in Figures 2-3 in main text, but we could not observe that in 5.6 nm thick samples. Actually, we measured several thin devices with thickness of WTe₂ ranging from 4.0 to 7.0 nm, we did not observe the hysteretic behavior in all the thin WTe₂ devices. The absence of spin momentum locking is a direct consequence of gradual vanishing of Weyl semimetal states and topological Fermi arc states in thinner WTe₂ samples.^{2,3}

***Comment 4:** 4) For these materials, there are bulk conducting states. Have the authors estimated the contributions from bulk conducting states?*

What are the carrier densities for surface and bulk states?

Reply: Thank you for your valuable comment and suggestion. We then calculated the carrier density for surface and bulk states.

From the data of Hall effect of WTe₂flake in Supplementary Figure 12a, we know the dominated bulk carrier in WTe₂flakes is *n*-type with a carrier density of $1.41(\pm 0.02) \times 10^{20} \text{ cm}^{-3}$. This carrier density in our WTe₂ flakes is almost twice that of the *n*-type carrier density of bulk WTe₂ reported previously.⁴ The higher bulk carrier density of electrons in our very thin plates of several tens nanometers can be understood as following. In the large, single crystalline WTe₂ bulk, both low density of electrons and holes contribute nearly equally to the electrical conduction for its semimetal characteristics.⁵ With decreasing the thickness of WTe₂ to the nm level, the Fermi level will shift upward near the Weyl point.² Consequently The dominated carrier

changes to n -type and much more electrons (could be doubled) will be involved into the electrical transport.

To estimate the surface carrier density, we need to calculate from the Weyl orbit quantum oscillation in Figure 1 shown in main text, which is consist of two topological Fermi arcs and two bulk chiral Landau levels. According to Figure 1b, the Frequency of Weyl orbit is ~ 82.0 T, we can obtain the carrier density of 2D surface states $n_{\text{SS}} = ef_{\text{sdH}} / h = 1.90 \times 10^{12} \text{ cm}^{-2}$.^{6,7}

According to our previous quantum oscillation analysis,² the Fermi velocity is $v_{\text{F}} = 3.1 \times 10^5 \text{ m/s}$, quantum scattering time is $\bar{\tau} = 1.6 \times 10^{-13} \text{ s}$, the length of Fermi arc is $k_{\text{F}} = 0.032 \text{ \AA}^{-1}$.² Therefore, the quantum mobility of 2D Fermi arc state can be calculated as $\mu = ev_{\text{F}}\bar{\tau} / \hbar k_{\text{F}} = 2520 \text{ cm}^2/\text{V}\cdot\text{s}$. Thus, the surface state conductance is $\sigma = en_{\text{2D}}\mu_{\text{2D}} = 0.766 \text{ mS}$, while the total conductance of WTe_2 at 2 K is $\sigma_{\text{total}} = 1/R_{\text{sheet}} = 42 \text{ mS}$. The ratio of surface conduction to total conduction is, therefore, $\alpha = \sigma_{\text{2D}} / \sigma_{\text{total}} = 1.8\%$, This rather low proportion of surface state conduction is reasonable, because topological Fermi arc states and high density bulk carrier coexist near the Fermi level ($n_{\text{SS}} = 1.90 \times 10^{12} \text{ cm}^{-2}$, and $n_{\text{bulk}} = 1.41 \times 10^{20} \text{ cm}^{-3}$). This case is opposite to the relative large surface state conduction in topological insulators.⁶ Nevertheless, the 2D surface conduction in WTe_2/Py can be obviously enhanced due to the formation of Rashba interface, as shown in Figures 1c and 1d in main text.

The discussion on surface state conduction and bulk conduction was added into the supplementary note 3.

***Comment 5:** 5) The technique to probe the spin-momentum locking was first used in Nat Nanotechnol 9, 218-224 (2014), but it was criticized by several later reports, including fringe field effect (E. K. de Vries, et*

al, Phys. Rev. B, 92, 201102 (2015)), and other spurious effects (Pengke Li and Ian Appelbaum, Phys. Rev. B 93, 220404(R) (2016)).

Have the authors ruled out those possibilities?

Reply: Thank you for your suggestion to exclude the fringe field effect and other spurious effect.

According to the work reported by E. K. de Vries, they could observe the voltage hysteresis loop not only for the case that the magnetization is perpendicular to the DC current but also for the one that magnetization is parallel to the DC current. They attributed to their observation to the fringe field induced by the asymmetry of these triangular features of Bi₂Se₃ grain edges.⁸ This asymmetric fringe magnetic field along z direction will also give rise to a Hall-like voltage loop, as shown in Figure R7. However, this case can be excluded as the origin in our case for following reasons.

First of all, topological insulator Bi₂Se₃ tends to grow as a triangle, as shown in Figure R8a. There are a great number of triangle shape grain edges, which will give rise to the fringe magnetic field when Bi₂Se₃ is next to ferromagnetic layer. However, our mechanically exfoliated WTe₂ is quite flat without this asymmetric edge. Figure R8b gives a typical optical image of our exfoliated WTe₂. Moreover, we have etched our WTe₂ into rectangle ribbons during the fabrication, and the region with terraces is removed by Ar during the device fabrication.

[Redacted: (Fig. 6) De Vries, E. *et al.* Towards the understanding of the origin of charge-current-induced spin voltage signals in the topological insulator Bi₂Se₃. *Phys. Rev. B* 92, 201102 (2015).]

Figure R7 The simulation of fringe field induced by triangle Bi_2Se_3 edge in PRB 92 201102.⁸

Figure R8(a) The atomic force microscope images of Bi_2Se_3 in PRB 92 201102.⁸ (b) typical optical image of WTe_2 flakes without step edges. The left panel is the fresh exfoliated WTe_2 , and the right panel is the final device for spin momentum locking. The scale bar is 10 μm .

More important, we could only observe the voltage hysteresis loop for $I//b$ $H//b$ (the topological Fermi arc state), but this feature is absent for $I//a$ $H//a$ (Figure 2h and Figures R1 and R4). If our observation was caused by the fringe field, we could have observed the voltage hysteresis loop for $I//a$ $H//a$ too. Therefore, the fringe field can be excluded as the origin in our experiments.

In Pengke Li's work (PRB 93, 220404R 2016), they observed a nonlocal voltage hysteresis in the topological trivial metal (Au) based heterojunctions. In this work, the author did not explain clearly the physical origin of the observations.⁹ Their voltage difference between two resistance states indeed shows significant temperature dependence, however, the anomalous Hall effect of CoFe should not exhibit so strong temperature dependence for the temperature below 50 K, because of the high Curie temperature of CoFe alloy. Therefore, they just rule out the spurious effect, such as, anomalous Hall effect of CoFe. Meanwhile, the strong temperature dependence in our observation (Figure 3f) also should exclude the possibility of anomalous Hall effect as the origin of our observations.

Actually, the signal observed in topological trivial metal Au based heterojunctions⁹ might originate from the bulk spin Hall effect or Rashba effect, because Rashba effect in Au is significantly strong¹⁰ and share the same spin texture as the surface state in topological insulators. The authors also mentioned these possible origins at the conclusion part.

In our work, to consolidate our conclusion, we make significant work on the spin orbit torques in WTe_2/Py heterojunctions to examine the spin accumulation in Fermi arc state. In the discussion part of main text, we excluded the Rashba effect as the origin of our enhanced spin conductivity, because of the very strong temperature dependence of spin conductivity for $I//b$. We also rule out the spin Hall effect which usually contributes to damping-like torque, because the enhanced spin conductivity points to the field like torque. Therefore, based on the above discussion, we believe that the spin momentum locking in WTe_2 is indeed related to the surface states and topological Fermi arc states.

References

- 1 Slonczewski, J. C. Conductance and exchange coupling of two ferromagnets separated by a tunneling barrier. *Phys Rev B***39**, 6995 (1989).
- 2 Li, P. *et al.* Evidence for topological type-II Weyl semimetal WTe₂. *Nature communications***8**, 2150 (2017).
- 3 Wang, Y. *et al.* Gate-tunable negative longitudinal magnetoresistance in the predicted type-II Weyl semimetal WTe₂. *Nature communications***7**, 13142 (2016).
- 4 Zhu, Z. *et al.* Quantum oscillations, thermoelectric coefficients, and the fermi surface of semimetallic WTe₂. *Physical review letters***114**, 176601 (2015).
- 5 Ali, M. N. *et al.* Large, non-saturating magnetoresistance in WTe₂. *Nature***514**, 205 (2014).
- 6 Tang, J. *et al.* Electrical detection of spin-polarized surface states conduction in (Bi_{0.53}Sb_{0.47})₂Te₃ topological insulator. *Nano letters***14**, 5423-5429 (2014).
- 7 Mallinson, R., Rayne, J. & Ure Jr, R. de haas-van alphen effect in n-type Bi₂Te₃. *Physical Review***175**, 1049 (1968).
- 8 De Vries, E. *et al.* Towards the understanding of the origin of charge-current-induced spin voltage signals in the topological insulator Bi₂Se₃. *Phys Rev B***92**, 201102 (2015).
- 9 Li, P. & Appelbaum, I. Interpreting current-induced spin polarization in topological insulator surface states. *Phys Rev B***93**, 220404 (2016).
- 10 LaShell, S., McDougall, B. & Jensen, E. Spin splitting of an Au (111) surface state band observed with angle resolved photoelectron spectroscopy. *Physical review letters***77**, 3419 (1996).

Reply to the comments of NCOMMS-18-15480-T

Response to Reviewer #2 :

Reviewer #2:

It is an interesting work and addresses a key topic. The field of Weyl semimetals has recently experienced a surge in all sorts of theoretical and experimental activities since the original proposal on the existence of Weyl points and Fermi arcs states in iridate materials. However, most work up to date has been done to detect Fermi arcs themselves and study chiral anomaly effect, two most striking features of Weyl semimetals, but the topological nature of the Fermi arcs states suggests their genuine applications in surface transport phenomena (as is the case of the topological insulators), something that is only beginning to appear. Therefore the present manuscript is important, timely, and potentially interesting to a broad physics community.

Comment 1:*However, the authors need to pay attention to the following points before the decision on publication is made: First, spin -momentum locking in topological insulators follows directly from the Hamiltonian of the surface state written in the form $H=v [k \cdot ez] \cdot \text{Sigma}$. There is no such Hamiltonian for the arc states, hence there is no spin momentum locking per se. On general ground, one can only expect that the spins along the arc should wind from one Weyl point to another Weyl point since the arc connect the points of opposite chirality. This has to be clearly explained in the manuscript and calling this effect “locking” is confusing.*

Reply: The referee asked a very good question. First of all, in materials with strong spin-orbit coupling (SOC), the spin and momentum degrees of freedom are coupled together. At surfaces, the inversion symmetry is broken, so the surfaces state would generally be spin-polarized with

the spin direction tied with the momentum k , showing a spin texture in k space. In our work and in literature, the terminology “spin-momentum locking” refers to this kind of behavior.

As the referee mentioned, this concept has been widely discussed for topological insulators (like Bi_2Se_3), where it can be explicitly manifested in a simple surface-state Hamiltonian (like $H=v [k \cdot e_z] \cdot \text{Sigma}$). In comparison, it is difficult to obtain a simple $k \cdot p$ type effective Hamiltonian for surface Fermi arcs in Weyl semimetal materials. This is because the Fermi arcs are generally extended in the surface Brillouin zone, rather than centered at a high-symmetry point, and their shapes are less constrained. But the absence of a simple surface-state model does not mean the surface states are not spin-momentum-locked. From first-principles calculations, we have explicitly shown that the Fermi arc surface states exhibit spin-momentum-locked spin textures [Fig. 2(b) of main text and Fig. 6 of the Supplemental Materials].

Indeed, the terminology “spin-momentum locking” has already been used in previous works on the Fermi arc states. For example, in the ARPES experiment [Science, 347 294-298 (2015)],¹ the authors directly image the spin-momentum locking features. They stated that (on page 297) *“The direction of spin polarization is reversed upon ..., which shows the spin-momentum locking property and the singly degenerate nature of the Fermi arc surface states”* Also, in the work Phys. Rev. B 92, 115428 (2015),² the authors stated in the abstract that they *“revealed the Fermi arcs with spin-momentum-locked spin texture”* by using first-principles calculations.

Thus, we think the usage of the terminology “spin-momentum locking” here is consistent with its usage in literature and is appropriate. Following the referee’s suggestion, in the revised manuscript, we have cited the two relevant references mentioned above, and added the following explanation to clarify the issue. Please find them in page 8 line 10.

“Since the Fermi arcs are usually extended in the surface Brillouin zone and their shapes are less constrained, the spin texture for the arc states generally cannot be described by a simple local Hamiltonian. However, the spin-momentum-locked spin textures can be revealed

by first-principles calculations and mapped out in experiment. Our calculation result for the Fermi arc spin texture is shown in Fig.2(b).”

***Comment 2:** Second, the spin accumulation in topological insulators was studied theoretically in great detail in a recent Nature Communications 7, 10878 (2016) where the transfer of spin through the bulk has been emphasized as the primary mechanism of the spin accumulation. The authors here assume that it is a single surface phenomenon but bulk boundary correspondence, the genuine feature of topological systems seems to be missing in such assumption. The authors should discuss this point in greater details.*

Reply: Thanks for the valuable comment. In the work [Nat. Comm. 7, 10878 (2016)] mentioned by the referee, the authors revealed that for topological insulators, the spin-polarized transport from the topological surface states can be understood as a result of the spin Hall effect in the bulk, establishing the bulk boundary correspondence. Since a Weyl semimetal may be regarded as the limiting case of a topological insulator with the gap approaching zero, we expect that the similar understanding also applies here. Indeed, with strong spin-orbit coupling and broken inversion symmetry, Weyl semimetals are expected to have large spin Hall conductivity in the bulk, which has been demonstrated in PRL 117, 146403 (2016). Thus, a bulk boundary correspondence should still exist, and the spin-polarized transport from the Fermi arc surface states may also be naturally connected to the spin Hall effect in the bulk: under an applied E field, the resulting transverse pure spin current in the bulk leads to opposite spin accumulations on the top and bottom surfaces, generating surface spin-polarized charge currents according to the Fermi arc spin textures. In [Nat. Comm. 7, 10878 (2016)], the unusual effect of the surface disorders on the spin relaxation has also been revealed from model calculations. We expect similar physics here but a quantitative study is beyond the scope of the current work.

We thank the referee again for the comment. In revision, we have cited the mentioned reference and added the following discussion in page 8 line 22.

“As revealed in Ref. 42 in main text, for topological insulators, the spin-polarized transport from the topological surface states can be understood as a result of the spin Hall effect in the bulk, establishing the bulk boundary correspondence. Given that a Weyl semimetal may be regarded as the limiting case of a topological insulator (with the gap approaching zero) and it usually exhibit strong spin Hall effect,³ a similar picture also applies here. Namely, under an applied E field, the resulting transverse pure spin current in the bulk of a Weyl semimetal leads to opposite spin accumulations on the top and bottom surfaces, generating surface spin-polarized charge currents according to the Fermi arc spin textures.”

***Comment 3:** Third, surface transport properties have been studied recently in a series of works[Phys. Rev. B 93, 235127 (2016), Phys. Rev. B 97, 085142 (2018)]. As the authors study a 2D transport in their system, and highlight rather high values for spin conductivity, it would be good to see a discussion in relation to the theoretical insights brought by these recent studies.*

Reply: We thank the referee for the valuable suggestion and for bringing into our attention these interesting theoretical works. In Phys. Rev. B 93, 235127 (2016), the authors found that the scattering between the Fermi arc surface states and bulk states for a Weyl semimetal leads to dissipation in the transport. In Phys. Rev. B 97, 085142 (2018), the authors found that such scattering strongly depends on the shape of the Fermi arc. The straight arc geometry is very disorder tolerant. These conclusions are quite interesting and relevant to our work.

Following the referee’s suggestion, in revision, we have cited the mentioned works and added the following discussion in page 19 line 5.

“It has been shown that several factors could affect the transport of the Fermi arc states. For example, the scattering between the Fermi arc surface states and bulk states for a Weyl semimetal would lead to dissipation in the transport.⁴ In addition, such scattering has sensitive dependence on the arc geometry.⁵ It has been found that straight arc geometry is very disorder tolerant. For WTe_2 studied here, its Fermi arc states coexist with the bulk states due to its type-II nature, and the arc shape is not quite straight. Hence, we expect the scattering effects would be important in the dissipative transport. However, these previous theories are about type-I Weyl semimetals. A theory on type-II Weyl materials would be desired in order to give more appropriate account of our experiment.”

References

- 1 Xu, S.-Y. *et al.* Observation of Fermi arc surface states in a topological metal. *Science* **347**, 294-298 (2015).
- 2 Sun, Y., Wu, S.-C. & Yan, B. Topological surface states and Fermi arcs of the noncentrosymmetric Weyl semimetals TaAs, TaP, NbAs, and NbP. *Phys Rev B* **92**, 115428 (2015).
- 3 Peng, X., Yang, Y., Singh, R. R., Savrasov, S. Y. & Yu, D. Spin generation via bulk spin current in three-dimensional topological insulators. *Nature communications* **7**, 10878 (2016).
- 4 Gorbar, E., Miransky, V., Shovkovy, I. & Sukhachov, P. Origin of dissipative Fermi arc transport in Weyl semimetals. *Phys Rev B* **93**, 235127 (2016).
- 5 Resta, G., Pi, S.-T., Wan, X. & Savrasov, S. Y. High surface conductivity of Fermi-arc electrons in Weyl semimetals. *Phys Rev B* **97**, 085142 (2018).

REVIEWERS' COMMENTS:

Reviewer #1 (Remarks to the Author):

After reading the response to my previous comments, I think this manuscript has been greatly improved by the authors. This manuscript reports a very interesting experimental observation and is recommended for publication in Nature Communications.

Reviewer #2 (Remarks to the Author):

The authors addressed my comments in full volume therefore I am happy to recommend the publication in the present form.